# DANAE^++^: A Smart Approach for Denoising Underwater Attitude Estimation [note 1]

**DOI:** 10.3390/s21041526

**Published:** 2021-02-22

**Authors:** Paolo Russo, Fabiana Di Ciaccio, Salvatore Troisi

**Affiliations:** 1Department of Computer, Control and Management Engineering “Antonio Rubert”, University of Rome La Sapienza, Via Ariosto 25, 00185 Rome, Italy; paolo.russo@diag.uniroma1.it; 2International PhD Programme “Environment, Resources and Sustainable Development”, Department of Science and Technology, Parthenope University of Naples, Centro Direzionale Isola C4, 80143 Naples, Italy; salvatore.troisi@uniparthenope.it

**Keywords:** attitude estimation, autoencoders, deep learning, denoising, Kalman filter, underwater environment

## Abstract

One of the main issues for the navigation of underwater robots consists in accurate vehicle positioning, which heavily depends on the orientation estimation phase. The systems employed to this end are affected by different noise typologies, mainly related to the sensors and to the irregular noise of the underwater environment. Filtering algorithms can reduce their effect if opportunely configured, but this process usually requires fine techniques and time. This paper presents DANAE${}^{++}$, an improved denoising autoencoder based on DANAE (deep Denoising AutoeNcoder for Attitude Estimation), which is able to recover Kalman Filter (KF) IMU/AHRS orientation estimations from any kind of noise, independently of its nature. This deep learning-based architecture already proved to be robust and reliable, but in its enhanced implementation significant improvements are obtained in terms of both results and performance. In fact, DANAE${}^{++}$ is able to denoise the three angles describing the attitude at the same time, and that is verified also using the estimations provided by an extended KF. Further tests could make this method suitable for real-time applications in navigation tasks.

## 1. Introduction

Localization is one of the most important tasks for unmanned robots, especially in underwater scenarios. Being a highly unstructured and GPS-denied environment, and being characterized by different noise sources and by the absence of man-made landmarks, the underwater setting provides more challenges for orientation estimation. In a typical configuration, the Euler angles representing the vehicle attitude (roll, pitch and yaw) are obtained through the integration of raw data acquired by the sensors embedded into an Inertial Measurement Unit (IMU), or in the more cost-effective Attitude and Heading Reference System (AHRS). One of the most successful methods to perform this elaboration is based on the Kalman Filter (KF) [1], in its linear and non-linear versions [2]. Although known as the perfect estimator under some assumptions, the estimation provided by the KF strongly depends on a good knowledge of the error covariance matrices describing the noise affecting the system. Moreover, the on-line computation of these matrices is often required for any time-varying or nonlinear system, squaring the number of necessary updating steps at each time step. Finally, the procedure employed to accurately fine-tune the filter parameters is known to be unintuitive, requiring specific settings for different scenarios [3].

In order to overcome these issues we present DANAE${}^{++}$, an improved version of DANAE [4], which is a deep denoising autoencoder developed to attenuate any source of error from the attitude estimation of linear and an extended KFs. In this improved version, DANAE${}^{++}$ gets as inputs the intermediate parameters provided by the filtering algorithms: this configuration proved to further reduce the total error affecting the final estimation. Extensive tests performed on two different datasets to evaluate the Euler angles showed the power of our approach, with a sensible improvement of both mean squared error and max deviation w.r.t. the ground truth data (GT). The strength of our proposed method consists in its ability to act as a full-noise compensation model for both noise and bias errors, without the need to separately process each influencing factor. The remainder of this study is organized as follows: Section 2 presents a brief literature review for Kalman-based algorithms and deep learning techniques applied on similar tasks. Section 3 introduces the theoretical concepts at the basis of our study, i.e., the orientation estimation process and the filtering techniques usually employed for the task, the U-Net model from which DANAE${}^{++}$ took inspiration and the general architecture of deep denoising autoencoders. Section 4 contains the characteristics of the datasets used for the experiments, followed by a concise description of our algorithms and of DANAE${}^{++}$ architecture. In Section 5 the results of all of the experiments are summarized and commented upon, with some final considerations regarding the topic and future possible improvements reported in Section 6.

## 2. Related Works

Robots’ performances are strongly affected by a correct pose and position determination, on which an accurate orientation estimation has a great impact, especially indoor or in underwater environments [5], where GPS support can frequently not be guaranteed. Since any kind of external factors combined with the sensors’ integration difficulties can lead to errors in the resulting angles, it is of fundamental importance to minimize error sources and their effects. The use of Kalman filtering techniques in robotic applications is ubiquitous. For example, the authors of [6] developed a GPS/IMU multisensor fusion algorithm to increase the reliability of the position information, while another interesting approach has been presented by [7], which uses KF to estimate the sensors’ signals’ noise or their biases. Over the last year, underwater navigation has seen a huge development of KF-based algorithms for orientation estimation; some interesting underwater applications have been developed by the author of [8,9]. Nowadays, small-scale robots usually mount more affordable sensor systems (e.g., AHRS), which greatly benefit from the power of KF and can equally provide high precision and reliable results. An example is the work of the authors of [10], who proposed an effective adaptive Kalman filter, which is able to exploit low-cost AHRS for efficient attitude estimation under various dynamic conditions, and of [11], who evaluated orientation estimation performances of smartphones in different settings.

The Linear Kalman Filter (LKF) takes as a basic assumption the linearity of the system dynamics formulation. When both state and observations are non-linear, extended and unscented KFs are used. Non-linear implementations of the filter were developed for example in [12], in which the robot pose is obtained by fusing camera and inertial data with an Extended Kalman Filter (EKF), and in [13] where the same task is accomplished using an Unscented Kalman Filter (UKF). For a detailed comparison between different Kalman filters, see [2] or [14].

Beside the critical task of sensor fusions, the estimation of both the sensors’ biases and noises is also crucial for an effective navigation system. Nonetheless, Kalman filtering-based techniques constitute a powerful approach even to solving this problem, allowing to estimate both the state and the sensors’ biases. For example, the authors of [15] exploited a KF for accurate biases estimation in a distributed-tracking system, while the authors of [16] identified and successfully removed noise using a KF in a real-time application. In order to reduce the noise and compensate for the drift of the Micro Electro Mechanical Systems (MEMS) gyroscope during usage, the authors of [17] proposed a Kalman filtering method based on information fusion. In [18], the authors proposed an algorithm based on an external acceleration compensation model to be used as a modifying parameter in adjusting the measurement noise covariance matrix of the EKF. In general, the noise matrices expressing the covariance of the measurements and of the process can be fine-tuned to reduce as much as possible the noise influence. However, this presents some drawbacks. An example is that the procedure requires on-line computation of the error covariance matrix for any time-varying or nonlinear system, squaring the number of necessary updating steps at each time step. Moreover, the formal nature of the Kalman filter makes the tuning phase a nonintuitive and complicated process [3].

The rise of Deep Learning has radically changed fields like computer vision and natural language processing. Since the spectacular success of ImageNet [19], Convolutional Neural Networks (CNNs) produce state-of-the-art accuracy on classification [20], detection [21] and segmentation [22] tasks, with Recurrent Neural Networks (RNNs) being the backbone models for speech recognition [23] and sequence generation [24]. Autoencoders [25] are another successful deep architecture where the aim is to reconstruct a signal by learning a latent representation from a set of data. They have been used for several tasks, as realistic text and image generation. One of the main models developed in this field is the U-Net [26], which performs effective semantic segmentation on medical images by exploiting skip connections on encoder–decoder layers. Variational Autoencoders [27] (VAEs) play an important role in text generation tasks, when semantically consistent latent space is needed; however, VAEs training generally suffers from mode collapse issues. The authors of [28] developed an autoencoder with binary latent space using a straight-through estimator: experiments showed that this approach maintains the main features of VAE, e.g., semantic consistency and good latent space coverage, while not suffering from the mode collapse, other than being much easier to train. One of the most successful uses of autoencoders is for noise removal. Since their introduction [29], Denoising Autoencoders (DAEs) have been used for a broad number of tasks, such as medical images improvement [30], speech enhancement [31] and electrocardiogram (ECG) signal boosting [32]. Unsupervised feature learning methods for single modalities (such as text or audio) have recently been developed; in [33], a deep denoising autoencoder was trained to predict the original clean audio feature from a deteriorated one, and then process the audio to conduct an isolated word recognition task. Traditional denoising methods, such as principal component analysis and dictionary learning, are computationally expensive on large datasets and not optimal for dealing with non-Gaussian noise. To overcome these issues, the authors of [34] applied state-of-the-art signal processing techniques to denoise gravitational wave signals embedded either in Gaussian or non-Gaussian noise, based on a sequence-to-sequence bi-directional Long-Short-Term-Memory (LSTM) RNN. In [35], a novel training procedure of an autoencoder network was proposed. In particular, a discriminator network was trained to distinguish between the output of the autoencoder and the data sampled from the training corpus. The autoencoder was then trained also by using the binary cross-entropy loss calculated at the output of the discriminator network. Expanding these concepts, we propose an intelligent deep denoising autoencoder to improve the Kalman filter outputs, significantly reducing the difficulties related to the parameters tuning, the biases definition and the effects of other noise sources. The strength of this method lies in the development of a full-noise compensation model, without the need to separately process each influencing factor.

## 3. Theoretical Notions and Method

In this section we provide some basic concepts regarding the attitude estimation process and the instrumentation used for the scope. A brief description of the linear and extended implementation of the KF and of the autoencoders will follow, and then our method will be discussed. It should be emphasized that DANAE${}^{++}$ is filter-agnostic and can be used seamlessly on linear and non-linear KFs as well as any other type of filter able to perform attitude estimation.

### 3.1. Orientation Estimation

The position and attitude of a body in 3D space can be defined by the three transnational and the three rotational coordinates, which relate the origin and orientation of the body-fixed coordinate system to the world frame. In particular, the orientation of a rigid body is usually expressed by a transformation matrix, the elements of which are generally parameterized in terms of Euler angles, rotation vectors, rotation matrices, and unit quaternions [36]. A detailed survey of this representation can be found in [37]. For the purposes of our study, some notions on reference systems and Eulerian and quaternion representations are given.

According to Euler’s theorem, any rotation can be described using the ϕ,θ,ψ angles or a rotational matrix *A*. The latter can be defined through the combination of the matrices D,C, and *B*: each of them describes the rotation around one of three axes X,Y, and *Z* in a specific order designated by the adopted convention (e.g., A=BCD). The Euler angles then represent the result of the three composed successive rotations, allowing to define the orientation of the body w.r.t. the local East-North-Up (ENU) or the North-East-Down (NED) coordinate frames.

The latter is mainly adopted when working with aerial [38] and underwater robots: in this case, the positive *X* axis points to the North, the positive *Y* axis to the East, and the positive *Z* axis follows the positive direction of the gravity force (down). Other custom body frames can be adopted when acquiring data from sensors, so it is of fundamental importance to specify this configuration in order to properly transform the measurements in the correct frame. With this state, the Euler angles are defined as follows:ϕ represents the rotation around the *X* axis, known as *roll*;θ defines the rotation around the *Y* axis, i.e., the *pitch* angle;ψ is related to the *yaw* angle around the *Z* axis.

As can be seen, Euler angles are intuitive and allow for a simple analysis of the body orientation in 3D space. However, they are limited by the *gimbal lock* phenomenon, which prevents them from measuring the correct angles when the *pitch* (θ) angle approaches $\pm 90^{\circ }$. Another issue related to the dynamics of rigid bodies is the *singularity* that can occur in the Euler angle parameterization. For a detailed discussion on the topics, see [39] or [40].

Quaternions provide an alternative representation technique that does not suffer from these problematics, although it is less intuitive than the previous one. A quaternion q can be seen as a generalization of complex numbers [41], formally written as in Equation (Equation 1):(1)$$q=q_{0}+q_{1}\tilde{i}+q_{2}\tilde{j}+q_{3}\tilde{k}=\left[\begin{array}{c} q_{0}\\ \tilde{q} \end{array}\right].$$

For the scope of this paper, we will only introduce some quaternion expressions that will be used in the extended KF implementation. A vector r can be rotated by θ degrees around the reference vector u using (Equation 2), where the rotation matrix C can be defined as in (Equation 3).
(2)$$r^{\prime }=Cr.$$
(3)$$C=\left[\begin{array}{ccc} q_{0}^{2}+q_{1}^{2}-q_{2}^{2}-q_{3}^{2} & 2(q_{1}q_{2}-q_{0}q_{3}) & 2(q_{1}q_{3}+q_{0}q_{2})\\ 2(q_{1}q_{2}+q_{0}q_{3}) & q_{0}^{2}-q_{1}^{2}+q_{2}^{2}-q_{3}^{2} & 2(q_{2}q_{3}-q_{0}q_{1})\\ 2(q_{1}q_{3}q_{0}q_{2}) & 2(q_{2}q_{3}+q_{0}q_{1}) & q_{0}^{2}-q_{1}^{2}-q_{2}^{2}+q_{3}^{2} \end{array}\right].$$

Finally, the first derivative of a quaternion is defined in (Equation 4), where w is the angular velocity in the X,Y, and *Z* directions. This equation will give us the possibility to directly use the gyroscope measures to transform the quaternion into a rotation matrix as in (Equation 3).
(4)$$\dot{q}=\frac{1}{2}q\times w.$$

### 3.2. Sensors’ Characteristics

As already stated, AHRS integrates the magnetometer to the basic IMU configuration containing a gyroscope and an accelerometer. The raw data acquired by the MEMS-AHRS sensors can have possible errors due to the system design, other than being affected by thermal and electronic-related noise, usually modeled as additive Gaussian noise. This entails deviations and oscillations around the correct value that can be reduced by prior calibration procedures [42]. MEMS contain limited size vibratory-rate gyroscopes, which have no rotating parts; this makes their installation easier and lowers their costs but at the same time, combined with even the slightest fabrication imperfections, leads to sensitivity issues that inevitably increase the noise levels in the angular velocity measurements [43]. Furthermore, another critical error is due to sensor drift, which theoretically makes the position error grow exponentially over time while it linearly increases for heading and velocity [44].

The accelerometer is very sensitive to vibrations and mechanical noise: this means that it does not solely measure gravity, but the result of many additional forces including the gravitational acceleration [45]. Moreover, MEMS accelerometers are characterized by lower accuracy than traditional high-performance ones [46].

Finally, the output of a magnetometer also depends on multiple factors, mainly related to offsets and sensitivity errors. Besides the instrumentation-related influences, magnetic field sensors suffer from magnetic perturbations. The presence of both ferromagnetic materials and electromagnetic systems heavily affects the measurements, causing artificial biases, scale factors and non-orthogonality errors which are very difficult to detect and compensate for [47].

Three main sources of attitude estimation errors can then be summarized as follows:Noise errors coming from the sensors’ noisy measurements;Bias errors deriving from wrong or missing calibration procedures;Filter errors due to a wrong or missing filter tuning procedure.

Generally speaking, the deterministic errors (static biases or scale factors) can be mathematically modeled, regardless of their constant or variable distribution over time. On the contrary, the random nature of stochastic errors implies that they can only be modeled as random variables characterized by some probabilistic distribution [16].

Some of the aforementioned sensor errors can be compensated through the integration of the three systems: combining this with proper sensor bias estimation procedures and opportune calibrations, an accurate orientation estimation can be obtained. Nevertheless, there are some noise sources that are difficult to detect and correctly remove with traditional methods. For this reason, we decided to develop DANAE${}^{++}$, a novel denoising autoencoder specifically trained to recognize and discard any kind of noise and disturbance from the KF estimations.

### 3.3. Kalman Filtering Techniques

The LKF is a widely used algorithm for the state estimation of dynamic systems since it is able to minimize the related variance under some perfect model assumptions (i.e., the expression of the process and measurement models as matrices and their related noise as additive Gaussian noise due to the linearity of the considered dynamic).

The system behavior in a discrete time setting can be described by a state Equation (Equation 5) and a measurement Equation (Equation 6):(5)$$x_{t}=Fx_{t-1}+Bu_{t-1}+w_{t-1}.$$
(6)$$z_{t}=Hx_{t}+v_{t}.$$
where $x_{t}$ is the state vector to be predicted, $x_{t-1}$ and $u_{t-1}$ are the state and the input vectors, respectively, at the previous time step and $z_{t}$ represents the measurement vector. *F* and *B* are the system matrices and *H* is the measurement matrix. The vectors $w_{t-1}$ and $v_{t}$ are respectively associated with the additive process noise and the measurement noise, assumed to be zero mean Gaussian processes. The final estimate is obtained by a first prediction step ((Equation 7) and (Equation 8)) followed by the update phase ((Equation 9)–(Equation 12)):(7)$${\hat{x}}_{t}^{-}=F{\hat{x}}_{t-1}^{+}+Bu_{t-1}.$$
(8)$$P_{t}^{-}=FP_{t-1}^{+}F^{T}+Q.$$
(9)$${\tilde{y}}_{t}=z_{t}-H{\hat{x}}_{t}^{-}.$$
(10)$$K_{t}=P_{t}^{-}H^{T}{\left(HP_{t}^{-}H^{T}+R\right)}^{-1}.$$
(11)$${\hat{x}}_{t}^{+}={\hat{x}}_{t}^{-}+K_{t}\tilde{y}.$$
(12)$$P_{t}^{+}=\left(I-K_{t}H\right)P_{t}^{-}.$$

The a posteriori state estimate ${\hat{x}}_{t}^{+}$ is obtained as a linear combination of the a priori estimate ${\hat{x}}_{t}^{-}$ and the weighted difference between the actual and the predicted measurements, the residual ${\tilde{y}}_{t}^{-}$ (see Equation (Equation 11)); the weight is defined by the Kalman gain (*K* in Equation (Equation 10)) and allows to minimize the a posteriori error covariance (*P* in Equation (Equation 12)) initially set by the user. Finally, *Q* and *R* are the covariance matrices of the process and of the measurement noise, respectively. *Q* models the dynamics uncertainty, and *R* represents the sensors internal noises. These matrices heavily affect the final filter performance, and thus a tricky tuning process is necessary to correctly estimate noises statistics. A proper fine-tuning is also important for sensors biases estimation; however, even in this case traditional approaches based on the KF suffer from implementation complexity and require non-intuitive tuning procedures [48]. In a non-linear dynamic system either the process or the measurement model cannot be defined with simple vectors and matrices’ multiplications. In this case, the EKF allows to efficiently deal with this issue by considering a model linearization around the current estimations. As the EKF is computationally cheaper than other nonlinear filtering methods (e.g., particle filter), it is widely used in various real-time applications, especially in the robotic and navigation fields. In this case, Equations (Equation 5) and (Equation 6) can be rewritten as:(13)$$x_{t}=f\left(x_{t-1},u_{t-1}\right)+w_{t-1}.$$
(14)$$z_{t}=h\left(x_{t}\right)+v_{t}.$$
where the matrices *F* and *H* have been replaced by f, the function that provides the current state $x_{t}$ on the basis of the previous state and control input, and by h, relating the current states to the measurements. These functions are processed at each time step to obtain the Jacobian matrix, first-order partial derivative of the function with respect to a vector, as described by Equations (Equation 15) and (Equation 16):(15)$$F_{t-1}=\frac{\partial f}{\partial x}{|}_{{\begin{array}{c} {\hat{x}}_{t-1}^{+},u \end{array}}_{t-1}}.$$
(16)$$H_{t}=\frac{\partial h}{\partial x}{|}_{\begin{array}{c} {\hat{x}}_{t}^{-} \end{array}}.$$

The estimation procedure is then similar to that of the LKF, with the main difference of obtaining the predicted estimations by the nonlinear functions in Equations (Equation 13) and (Equation 14).

### 3.4. Denoising Autoencoders

A DAE is a deep convolutional model that is able to recover clean, undistorted output starting from partially corrupted data as input. In the original implementation, the input data are intentionally corrupted through a stochastic mapping (Equation (Equation 17)):(17)$$\tilde{x}∼q_{D}\left(\tilde{x}|x\right).$$

Then, the corrupted input is mapped into a hidden representation as in the case of a standard autoencoder (Equation (Equation 18)):(18)$$h=f_{\theta }\left(\tilde{x}\right)=s\left(W\tilde{x}+b\right).$$

Finally, the hidden representation is mapped back to a reconstructed signal (Equation (Equation 19)):(19)$$\hat{x}=g_{\theta^{\prime }}\left(h\right).$$

During the training procedure, the output signal is compared with a reference signal in order to minimize the L2 reconstruction error (Equation (Equation 20)):(20)$$L\left(x-\hat{x}\right)=||x-\hat{x}{\left|\right|}^{2}=||x-s\left(W\tilde{x}+b\right){\left|\right|}^{2}.$$

### 3.5. U-Net Architecture

The U-Net [26] is a fully convolutional network originally developed for effective medical images analysis. It is able to achieve robust and accurate performance in several tasks like pancreas, brain tumor and abdominal computed tomography semantic segmentation [49,50,51]. Its architecture resembles the encoder–decoder model: a *contracting path* reduces the input data up to a set of high-level features, and an *expansive path* that upsamples the features back to the original size by exploiting *transposed convolutions* [52]. Encoder and decoder paths are linked by *skip connections* so that the $l_{d}^{i}$ layer of the decoder network receives as input both the feature maps from the $l_{d}^{i-1}$ decoder layer and the features map from the $l_{e}^{i-1}$ encoder layer. The presence of these long, symmetric shortcuts both reduce the vanishing gradient issue and improve the ability of the model to capture fine-grained details [53].

### 3.6. DANAE${}^{++}$

DANAE${}^{++}$ is a deep denoising autoencoder developed to recover the orientation estimation of robots and low-cost sensors from any kind of disturbance, considering the internal AHRS noise as well as that introduced by the filtering process. In fact, in this work DANAE${}^{++}$ has been tested on both linear and extended Kalman filters, but it can run on any kind of algorithm employed for the scope. The proposed architecture is inspired by WaveNet [54], which is a 1-dimensional U-Net model originally created for raw audio waveform generation; DANAE${}^{++}$ takes as input the roll, pitch and yaw estimations provided by the filter and produces as output the same angles recovered from the noise. DANAE${}^{++}$ can work with any input signal length, here denoted by *N*; without loss of generality, we performed our experiments using N=20.

To increase the generation ability of the architecture, DANAE${}^{++}$ has been further improved from its original structure by receiving as additional input the intermediate angles estimation calculated inside the filtering loop. Moreover, DANAE${}^{++}$ is able to estimate the three angles at the same time. For these reasons, the input dimension becomes MxN, where M=9 is the sum of the three estimated angles and the six intermediate ones extrapolated from the KF. As shown in Section 5, the aforementioned changes increase the final accuracy.

As regards the network structure, the encoder part of DANAE${}^{++}$ is made up of four *dilated* 1D convolutions, which bring the M×N input signal to a hidden representation made of 128 features. The decoder part transforms this representation to a 3×N output by alternating three transposed-dilated 1D convolutions to four standard ones (Figure 1). While the transposed convolution is exploited to increase the input resolution back to the original size, the ${\left(i+1\right)}^{th}$ standard convolution works on the sum of the $i^{th}$ encoder and the $i^{th}$ decoder outputs. This approach, loosely inspired by the WaveNet architecture [55], is able to enforce additional constrains on the encoder–decoder pipeline, enabling a more faithful signal reconstruction.

In our implementation, DANAE${}^{++}$ takes as input the noisy angles’ prediction performed by the LKF or EKF, together with the intermediate estimations obtained before their integration (i.e., the Euler angles analytically derived using the accelerometer/magnetometer and the gyroscope measurements), and as reference signal the ground truth angles provided by the dataset. It adds white noise during the training and tries to output the undistorted signal, forcing the network to recognize and discard any kind of disturbance.

For this reason, we underline that our method is able to remove both stochastic errors (e.g., electromagnetic- and thermo-mechanical-related ones) and systematic errors (due for example to sensor misalignment).

## 4. Experimental Setup

In this section the two datasets on which DANAE${}^{++}$ has been developed will be presented, outlying the measurements and the ground truth acquisition methodologies. A detailed overview of our experimental setup will then be given, describing the estimation acquisition–training–validation–testing pipeline of DANAE${}^{++}$ as well as the determination of the model hyper-parameters and settings.

### 4.1. Datasets

DANAE${}^{++}$ was originally conceived as a method to improve underwater positioning operations, with the aim of reducing the noise effects on the measurements acquired in this particularly unstructured environment. However, this setting poses strong challenges also regarding the acquisition of reliable GT data, which leads to a scarcity of available underwater datasets suitable for Deep Learning applications. For this reason, we decided to test DANAE${}^{++}$ on two public available datasets, acquired in both terrestrial (indoor and outdoor) and underwater environments. In this way, we also highlight the strength of our method, which produces remarkable improvements regardless of the working conditions.

The chosen datasets are the Oxford Inertial Odometry Dataset (OxIOD) [56] and the Underwater Caves Sonar Dataset (UCSD) [57].

OxIOD has been chosen for its accurate ground truth measurements over big heterogeneous settings. Developed for Deep Learning-based inertial odometry navigation, OxIOD provides 158 sequences (for a total of 42.587 km) of inertial and magnetic field data acquired from low-cost sensors. Five users made indoor and outdoor acquisitions while normally walking with their phone in hand, pocket or handbag and slowly walking, running and performing mixed motion modes. Different smartphones have been used to acquire the data, but its majority has been collected by an iPhone 7 Plus equipped with an InvenSense ICM20600. The gyroscope noise is $4mdps/\sqrt{\left(Hz\right)}$, with a sensitivity error of 1%, while the accelerometer noise is $100g/\sqrt{\left(Hz\right)}$. The three-axis geomagnetic sensor in the iPhone 7 Plus (Alps HSCDTD004A) has a measurement range of ±1.2 mT and an output resolution of 0.3T/LSB. A Vicon motion capture system was used to obtain the ground truth, provided only for the position with a precision down to 0.5 mm [56]. The UCSD has been collected by a Sparus Autonomous Underwater Vehicle (AUV) navigating in the underwater cave complex “Coves de Cala Viuda” in Spain. The vehicle explored two tunnels, following a 500 m-long path at a depth of approximately 20 m. Among the equipped sensors (e.g., DVL, sonar, etc.), a standard low-cost Xsens MTi AHRS and an Analog Devices ADIS16480 were mounted. The latter is a 10 DOF MEMS that provides more accurate raw sensor measurements and dynamic orientation outputs (obtained by their EKF fusion). Table 1 provides Sparus XSens MTi and ADIS AHRSs specifications. The elaboration of images containing six traffic cones placed on the seabed allowed to obtain the relative positioning of the vehicle. Unfortunately, the ground truth thus obtained is synchronized with the low-rate camera acquisitions, making the comparison with the high-rate IMU measurements inconsistent. For this reason, we assumed that the orientation directly provided by the AHRS could at first glance substitute the true ground truth. Despite not being a proper solution to the issue, this choice allowed us to understand the ability of DANAE${}^{++}$ to work in a true underwater scenario with its unique features.

### 4.2. Experiments

Some details on the experiments will be given in this section. Both datasets have been split into training and test sets; in the case of OxIOD, we used for each setting run 1 as a test set, leaving all of the other sessions as a training set.

UCSD instead provides a single file for each system containing all of the measurements stored during the entire survey. We then decided to split the data, using the first 80% to train DANAE${}^{++}$ and the remaining 20% to test the performances.

Three main phases can be distinguished: during the first one, the inertial and magnetic field data are integrated with a linear or extended KF, providing the estimation of the three Euler angles. In the second phase, these outputs are fed to DANAE${}^{++}$ for training, while in the third phase tests are performed using a pipeline of KF and DANAE${}^{++}$(Figure 2). All of the hyper-parameter values were found using the OxIOD *handheld* data set as a validation set. We empirically found that these values generalize well on both the OxIO and on UCS datasets. The network weights after training have been saved for later use, e.g., for model deployment on a robot. The code was developed in Python 3.6.9 running on Ubuntu 18.04, with the help of the Pytorch framework.

#### 4.2.1. Kalman Filters Initialization

We implemented the LKF in its most basic formulation following the equations from (Equation 7) to (Equation 12). The covariance matrices P, Q and R were initialized as an identity matrix, and no tuning has been done with relation to both the internal system and the measurements noises. This choice was made to highlight the ability of DANAE${}^{++}$ to denoise the estimations provided by the filter independently of its poor or erroneous initialization and/or tuning. This becomes particularly useful in those situations in which the finalization of those procedures is difficult or not possible.

The elaboration of the accelerometer and magnetometer raw data provided the measurements vector (see $Hx_{t}$ in Equation (Equation 6)), while the gyroscope-derived angles have been set as external input (see $Bx_{t-1}$ in Equation (Equation 5)).

Different procedures have been followed for the EKF. The filter logic is of course the same as the LKF, but as the linearizations and the use of quaternions is not very intuitive, a concise explanation of the implementation is reported here. We used the first-order linearized model to discretize and easily insert the system in our code: following Equation (Equation 4), and considering Equation (Equation 21) (where dt is approximated by calculating the difference of timesteps between samples at time *t* and t+1) we obtained Equation (Equation 22):(21)$${\dot{q}}_{t}=\frac{q_{t+1}-q_{t}}{dt}.$$
(22)$$q_{t+1}=\frac{dt}{2}S\left(w\right)q_{t}-\frac{dt}{2}Sb_{g}q_{t}+q_{t}.$$
where $b_{g}$ is the gyroscope bias in its three components along the *x*, *y* and *z* axes, and *S* is the skew-symmetric matrix equivalent to the cross-product. For a more detailed explanation, see [58]. After some calculations and following the LKF structure as in Equations (Equation 7)–(Equation 12), the EKF can then be described using Equation (Equation 23):(23)$${\left[\begin{array}{c} q\\ b_{g} \end{array}\right]}_{\begin{array}{c} t \end{array}}={\left[\begin{array}{cc} I_{4x4} & -\frac{dt}{2}S\left(q\right)\\ 0_{3x4} & I_{3x3} \end{array}\right]}_{\begin{array}{c} t-1 \end{array}}{\left[\begin{array}{c} q\\ b_{g} \end{array}\right]}_{\begin{array}{c} t-1 \end{array}}+{\left[\begin{array}{c} \frac{dt}{2}S\left(q\right)\\ 0_{3x3} \end{array}\right]}_{\begin{array}{c} t-1 \end{array}}w_{t-1}.$$

Once again, remember that *w* is the angular velocity vector in the three directions. The covariance matrices (*P*, *Q* and *R*) have been initialized again as identity matrices, avoiding any kind of tuning. The matrix *C* used to convert the filters states to the measured variables (see Equation (Equation 6)) is associated with the accelerations and magnetic field values, as shown in Equation (Equation 24):(24)$${\left[\begin{array}{c} \hat{a}\\ \hat{m} \end{array}\right]}_{\begin{array}{c} t \end{array}}=\left[\begin{array}{cc} C_{a} & 0_{3x3}\\ C_{m} & 0_{3x3} \end{array}\right]{\left[\begin{array}{c} q\\ b_{g} \end{array}\right]}_{\begin{array}{c} t \end{array}}.$$
where $C_{a}$ and $C_{m}$ are the matrices associated with the accelerometer and magnetometer, respectively. Please note that to simplify the reading, the vectorization of some variables has been omitted.

We assumed that the accelerometer gives an accurate reference in the vertical plane (*Z* axis) while the magnetometer is accurate in providing the reference in the horizontal plane, in particular in the magnetic north direction (*Y* axis). As the latter is more susceptible to external disturbance factors, we made a prior calibration of its measures [59], which, in the case of the OxIO dataset, led to an improvement of the EKF estimations.

#### 4.2.2. DANAE${}^{++}$ Setting

The layers have 128 3×3 kernels with an appropriate dilation value depending on the layer depth, while stride and padding have been fixed to 1. The Adam optimizer chosen for the training was set with a fixed learning rate of 0.002 with a batch size of 16. The number of epochs was set to 100 for UCSD and to 1 for each set of OxIOD. Additional experiments performed with different hyper-parameter values did not produce any sensible difference in the final accuracy, demonstrating the robustness of our approach.

## 5. Results

To numerically evaluate the performances of DANAE${}^{++}$, simple estimators such as mean deviation, maximum deviation and RMSE have been calculated with respect to the GT and compared with those of both the LKF and EKF. For the sake of brevity, we will include the images of DANAE${}^{++}$ tested on the EKF alongside with those provided by its previous implementation (DANAE, see Figure 3 and Figure 4). Table 2 and Table 3 report a detailed analysis of the results obtained with the OxIO dataset, while Table 4 and Table 5 report those for the UCS dataset.

The numerical values demonstrate that DANAE${}^{++}$ is able to considerably improve the performances on all estimators; this is valid for the LKF as well as on the EKF and for all the three angles on both datasets. Despite the strong noise affecting the KF predictions, DANAE${}^{++}$ is able to produce a sensible lowering of the mean deviation w.r.t the GT, upholding its strong denoising capability.

More in detail, DANAE${}^{++}$ results on the OxIO dataset produced a mean LKF RMSE reduction of 63%, ranging from 58% (ψ) to 67% (θ), and of 52% for the EKF, with a minimum of 25% (ψ) and a maximum of 60% (ϕ). Figure 5 shows the difference between the EKF and DANAE${}^{++}$ on the estimation of ϕ.

A similar result was found in the UCSD experiments: DANAE${}^{++}$ output faithfully resembles the reference signal for the estimated angles, reducing the LKF RMSE to a range between 57% and 60%. For the EKF, the reduction is instead between 54% (θ) and 61 (ϕ): Figure 6 reports the corresponding results for the θ angle. Unfortunately, ψ exhibits a perturbed behavior in both the estimated and ground truth values, which is the reason numerical values are omitted here. This can be probably related to erroneous sensors calibrations or to magnetometer effects, whose non-linearity results in a scale factor error. Moreover, electromagnetic-produced deviations can considerably alter the estimations of this angle [47].

It should be emphasized that DANAE${}^{++}$ works simultaneously on the three angles; this reduces the overall time consumption of ∼66%, thus proving to be a smarter solution than the previous version.

With the aim of further validating our method, we also compared the results of DANAE${}^{++}$ to those obtained applying a low-pass filter to the KF estimations. In particular, we implemented *Butterworth* (Scipy Butterworth filter: https://docs.scipy.org/doc/scipy/reference/generated/scipy.signal.butter.html) and a *cumulative* (Scipy cumulative filter: https://docs.scipy.org/doc/scipy/reference/generated/scipy.ndimage.uniform_filter1d.html) filters, both using the Python library *Scipy*.

Figure 7 compares the GT to the EKF estimations provided by DANAE${}^{++}$ and the Butterworth and cumulative filters w.r.t the GT. Results in Table 6 confirm that the denoising effect of our model outperforms that provided by both filters. This is because DANAE${}^{++}$ does not work on a specific frequency noise, but on any kind of influencing factor or bias regardless of their nature.

## 6. Conclusions

This paper presents DANAE${}^{++}$, an enhanced implementation of the previously developed deep denoising autoencoder for attitude estimation, DANAE. Despite the exceptional results obtained by the scientific community, attitude estimation is still considered a challenging task. This is particularly evident in complex scenarios such as those underwater, where different noise sources, unstructured settings and the absence of GPS heavily affect the orientation and positioning accuracy of the vehicles. The filtering algorithms employed to determine the Euler angles of roll, pitch and yaw are able to give state-of-the-art results through the integration of measurements provided by the gyroscope, accelerometer and magnetometer embedded in high-performing systems or in the cheaper but equally effective MEMS sensors. However, these filters generally require fine-tuning procedures, which constitute a non-trivial task, and can suffer from the effect of different disturbing factors and other internal and external noise sources, which are not easily detectable.

By leveraging the potential of recent progress in the Deep Learning field, we developed a denoising autoencoder that is able to recover attitude estimation signals from any kind of noise, thus attenuating the aforementioned issues’ impact. The DANAE${}^{++}$ architecture is loosely inspired by the U-Net and WaveNet models: it has an encoder part that contracts the signal to a set of high-level features through 1D convolutions, and a decoder part that upsamples them to the original size by exploiting transposed convolutions. Both paths are linked through skip connections, with the aim of reducing the vanishing gradient issue while improving the model’s ability to capture details.

DANAE${}^{++}$ was developed and tested on two datasets: the Oxford Inertial Odometry Dataset, acquired with low-cost sensors in different settings, and the Underwater Caves Sonar Dataset, collected by a Sparus Autonomous Underwater Vehicle. For each of them, training and a testing sets were defined. During the training, the network took as input the noisy angles estimations provided by the filters (LKF and EKF in our case) and the ground truth values provided by the datasets. One of the subsets of the OxIO dataset has been used to validate the model, empirically finding that the thus derived hyperparameters generalized well on the UCSD too. These network weights have been saved for later use, e.g., for a possible deployment of the model on a robot, in real time.

At the end of the test phase, an analysis of the performances was made: the orientation obtained by DANAE${}^{++}$ was evaluated through mean and maximum deviation and RMSE w.r.t. the GT. The results confirm that DANAE${}^{++}$ is able to improve the final estimations, providing a general reduction of the RMSE of more than 50% for both the datasets, independently of the used filter.

DANAE${}^{++}$ adds to its previous configuration some remarkable improvements that can be summarized as follows:In addition to the estimations provided by the filter, it takes as an input the intermediate attitude values analytically derived inside the filter loop from the sensors’ measurements; this solution was proven to increase the accuracy of the final results;Differently from its previous implementation, it is capable of denoising the three orientation angles at the same time, thus reducing the overall time consumption of ∼66%.

We emphasize that our method is capable of removing both stochastic errors (e.g., electromagnetic- and thermo-mechanical-related ones), and systematic errors (for example due to sensors misalignment), and that it is completely filter-agnostic. It is important to highlight that DANAE${}^{++}$ can work on the estimations provided by any kind of filter regardless of its initial configuration and tuning. In fact, in some situations it can be difficult or not even possible to successfully complete these procedures (e.g., when sensor characteristics are not known or when a lack of time does not allow it), so it could be useful to rely on a method that does not need to perform these operations. Considering the obtained numerical results and the aforementioned characteristics, DANAE${}^{++}$ proves to be a smarter solution than its previous version. We are trying to enhance the reliability of systems orientation, whose accuracy is strictly related to the final position determination, by merging classical methods with Deep Learning novelties. This powerful approach will be further enhanced, analyzing the possibility to work with the raw measurements acquired by the sensors and to further optimize the architecture. Moreover, experiments on real robot acquisitions will be made, and deployments for on-line applications will be investigated and tested.

## Figures and Tables

**Figure 1 sensors-21-01526-f001:**
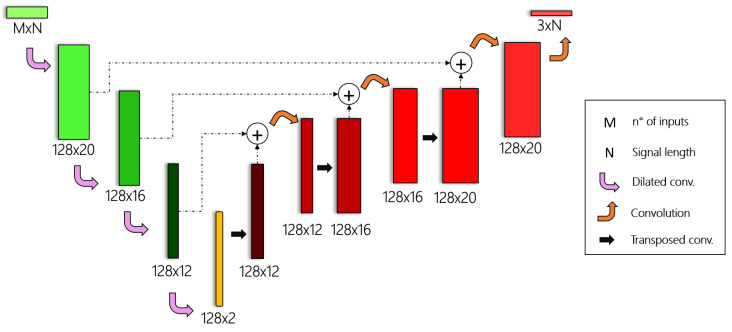
DANAE${}^{++}$ architecture: the dilated convolutions (green blocks) represent the encoder part of the model, while the transposed and standard convolutions (red blocks) constitute the decoder part.

**Figure 2 sensors-21-01526-f002:**
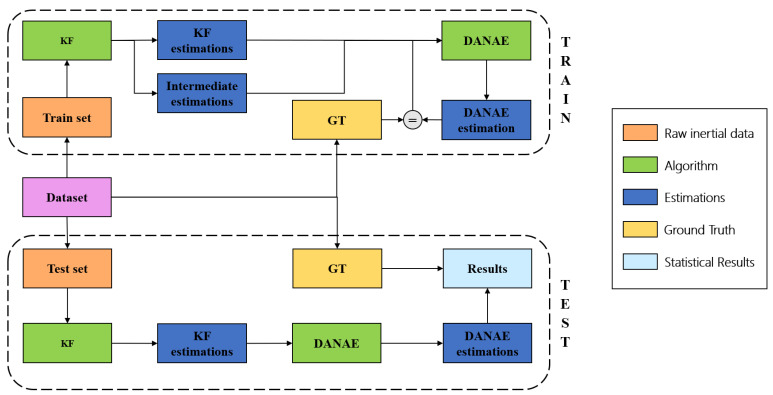
Workflow of the experiments: the upper section summarizes the training phase, while the bottom section represents the relative testing phase.

**Figure 3 sensors-21-01526-f003:**
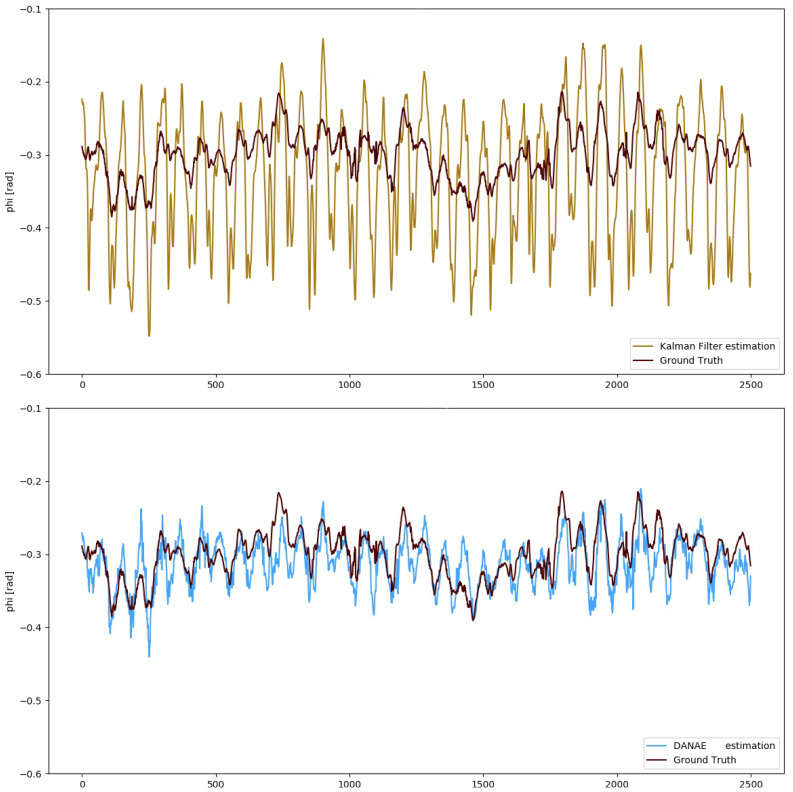
OxIO Dataset: roll angle estimation provided by the LKF (top, light brown) and DANAE (bottom, light blue) compared to the GT (dark red). This experiment was performed on a subsection of the slow walking set.

**Figure 4 sensors-21-01526-f004:**
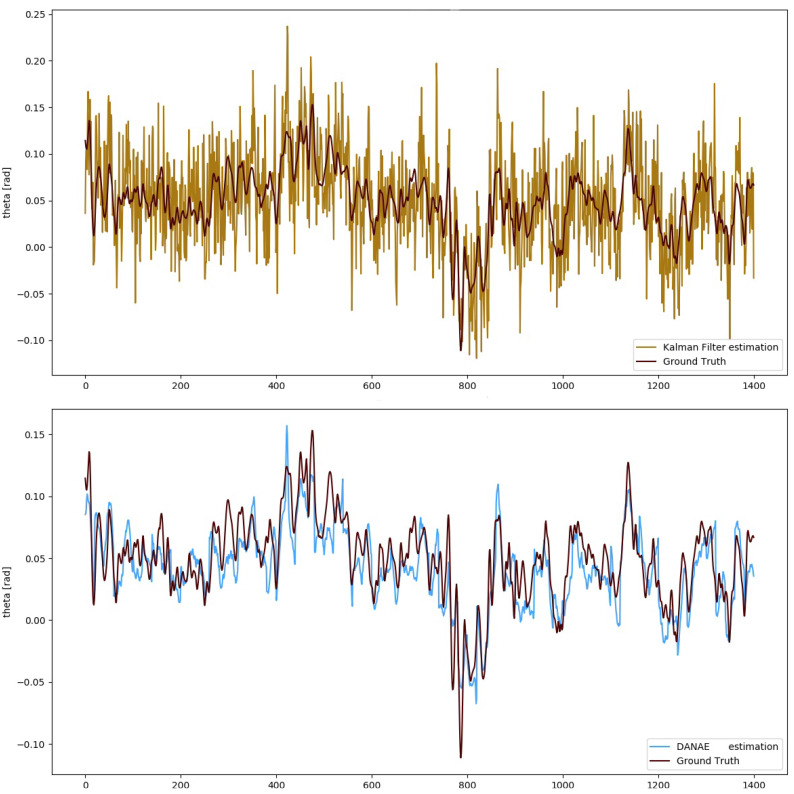
UCS Dataset: theta angle estimation provided by the LKF (top, light brown) and DANAE (bottom, light blue) compared to the GT (dark red).

**Figure 5 sensors-21-01526-f005:**
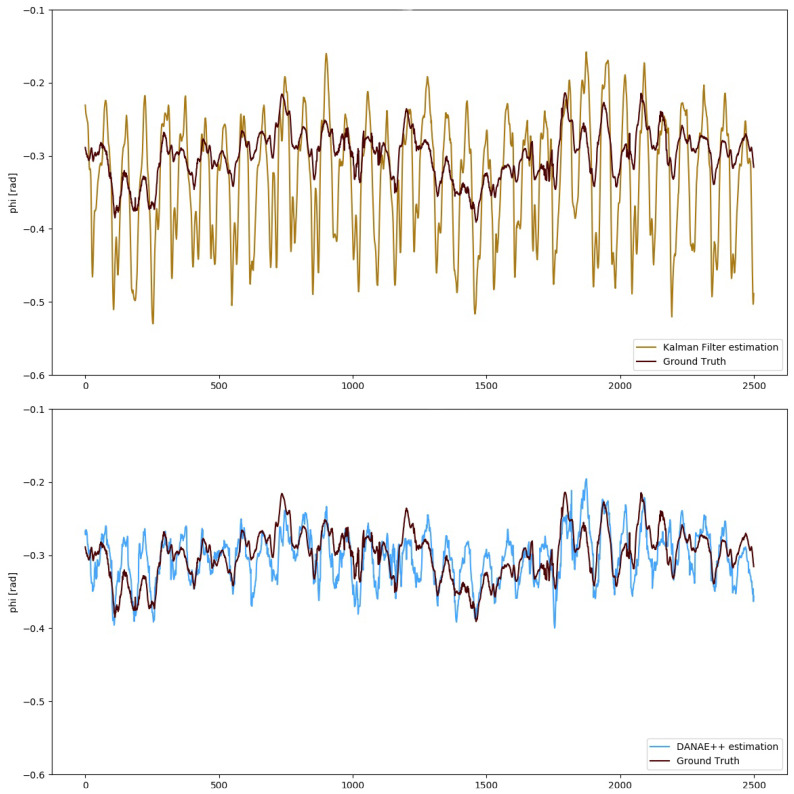
OxIO Dataset: roll angle estimation provided by the EKF (top, light brown) and DANAE${}^{++}$(bottom, light blue) compared to the GT (dark red). This experiment is made on a subsection of the slow walking set.

**Figure 6 sensors-21-01526-f006:**
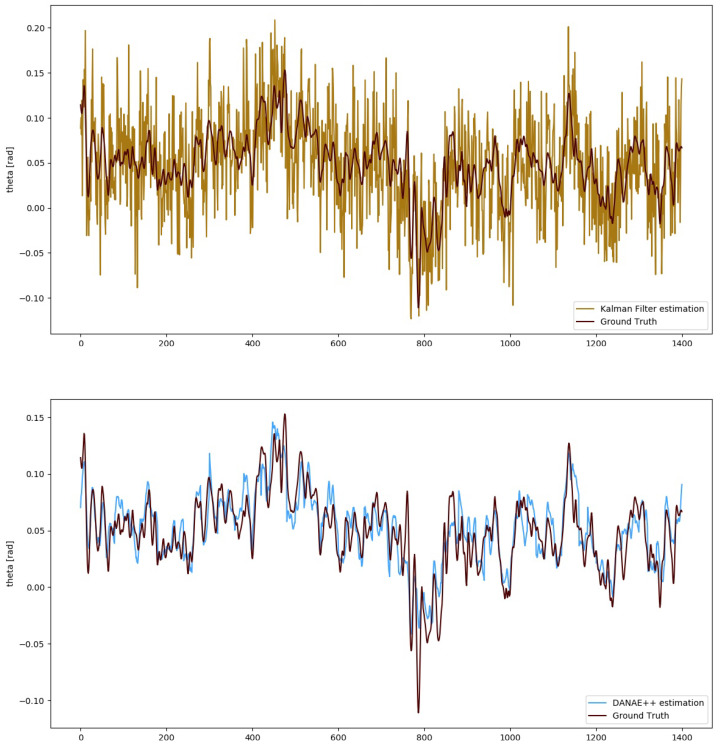
UCS Dataset: theta angle estimation provided by the EKF (top, light brown) and DANAE${}^{++}$(bottom, light blue) compared to the GT (dark red). This experiment was performed on a subsection of the slow walking set.

**Figure 7 sensors-21-01526-f007:**
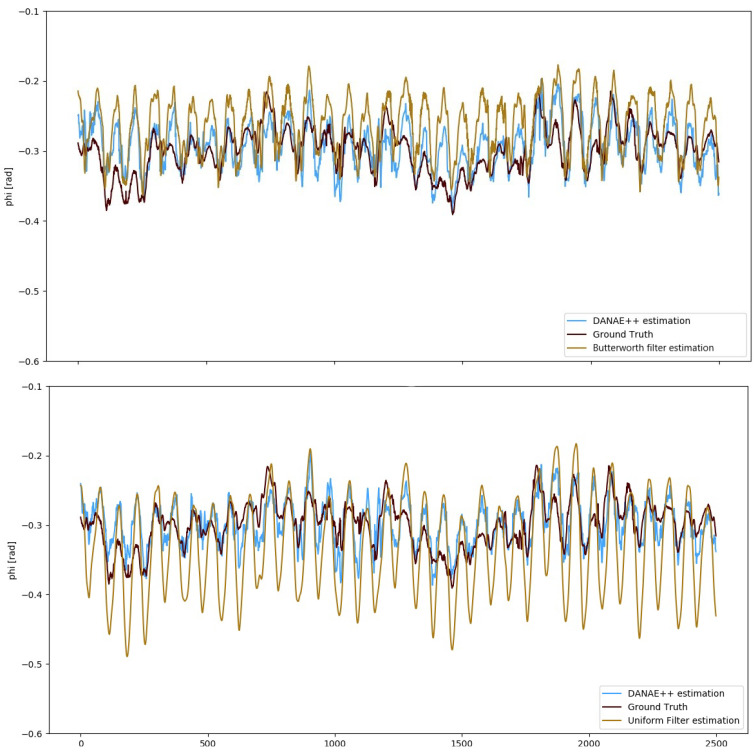
OxIO Dataset: roll angle estimation provided by DANAE${}^{++}$(light blue) compared to the butterworth (top) and the uniform1d (bottom) filters applied on the EKF outputs (light brown). The GT (dark red) is reported as reference in both the images.

**Table 1 sensors-21-01526-t001:** Sparus AUV AHRS specifications.

XSens MTi		ADIS16480	
Angular resolution	0.05 deg	Static accuracy (roll/pitch)	0.1 deg
Repeatability	0.2 deg	Static accuracy (heading)	0.3 deg
Static accuracy (roll/pitch)	0.5 deg	Dynamic accuracy (roll/pitch)	0.3 deg
Static accuracy (heading)	1 deg	Dynamic accuracy (heading)	0.5 deg
Dynamic accuracy	2 deg RMS		

**Table 2 sensors-21-01526-t002:** OxIO Dataset: evaluation of the performances of the LKF, DANAE and DANAE${}^{++}$ w.r.t. the GT for the three Euler angles.

	LKF			DANAE			DANAE${}^{++}$		
	ϕ	θ	ψ	ϕ	θ	ψ	ϕ	θ	ψ
Mean dev. [rad]	0.0661	0.0483	1.9518	0.0224	0.0157	0.7392	0.0237	0.0157	0.5756
Max dev. [rad]	0.2929	0.2134	2.7313	0.1382	0.1082	0.4925	0.1396	0.1064	0.1285
RMSE	0.0815	0.0600	2.4000	0.0282	0.0196	1.3194	0.0296	0.0197	1.0014

**Table 3 sensors-21-01526-t003:** OxIO Dataset: evaluation of the performances of the EKF, DANAE and DANAE${}^{++}$ w.r.t. the GT for the three Euler angles.

	LKF			DANAE			DANAE${}^{++}$		
	ϕ	θ	ψ	ϕ	θ	ψ	ϕ	θ	ψ
Mean dev. [rad]	0.0614	0.0485	0.4535	0.0216	0.0150	0.3636	0.0240	0.0149	0.2790
Max dev. [rad]	0.2724	0.2113	0.0189	0.1198	0.1100	0.7921	0.1632	0.1014	0.2482
RMSE	0.0762	0.0601	1.0478	0.0270	0.0187	0.8218	0.0301	0.0188	0.7860

**Table 4 sensors-21-01526-t004:** UCS Dataset: evaluation of the performances of the LKF, DANAE and DANAE${}^{++}$ w.r.t. the GT for the three Euler angles. Since the GT values of ψ are not reliable, the corresponding results are not reported here.

	LKF			DANAE			DANAE${}^{++}$		
	ϕ	θ	ψ	ϕ	θ	ψ	ϕ	θ	ψ
Mean dev. [rad]	0.0326	0.0328	-	0.0139	0.0147	-	0.0127	0.0142	-
Max dev. [rad]	0.1476	0.1751	-	0.0671	0.0769	-	0.0616	0.0712	-
RMSE	0.0410	0.0412	-	0.0177	0.0190	-	0.0162	0.0184	-

**Table 5 sensors-21-01526-t005:** UCS Dataset: evaluation of the performances of the EKF, DANAE and DANAE${}^{++}$ w.r.t. the GT for the three Euler angles. Since the GT values of ψ are not reliable, the corresponding results are not reported here.

	EKF			DANAE			DANAE${}^{++}$		
	ϕ	θ	ψ	ϕ	θ	ψ	ϕ	θ	ψ
Mean dev. [rad]	0.0249	0.0341	-	0.0125	0.0141	-	0.0126	0.0140	-
Max dev. [rad]	0.1382	0.1578	-	0.0807	0.0882	-	0.0616	0.0824	-
RMSE	0.0427	0.0412	-	0.0163	0.0180	-	0.0160	0.0179	-

**Table 6 sensors-21-01526-t006:** OxIO Dataset: evaluation of the performances of DANAE${}^{++}$, butterworth and uniform1d filters w.r.t. the GT for the three Euler angles.

	DANAE${}^{++}$			Butterworth Filter			Uniform1d Filter		
	ϕ	θ	ψ	ϕ	θ	ψ	ϕ	θ	ψ
Mean dev. [rad]	0.0240	0.0149	0.2790	0.0470	0.0204	1.0113	0.0535	0.0456	0.4537
Max dev. [rad]	0.1632	0.1014	0.2482	0.1488	0.1012	2.1764	0.1841	0.1880	0.0926
RMSE	0.0301	0.0188	0.7860	0.0547	0.0254	1.3377	0.0640	0.0561	1.0039

## Data Availability

Publicly available datasets were analyzed in this study. This data can be found here: OxIO Dataset: http://deepio.cs.ox.ac.uk/ (accessed on 12 December 2020); UCS Dataset: https://cirs.udg.edu/caves-dataset/ (accessed on 12 December 2020).

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
