# Peer review of "DANAE++: A Smart Approach for Denoising Underwater Attitude Estimation†"

_sensors, 2021, doi:10.3390/s21041526_

Round 1

Reviewer 1 Report

The paper has an original, scientific character, related to a smart approach for denoising underwater attitude estimation.
The content of the article is consistent with the scientific area of the journal Sensors. The subject raised by the authors is current and so far rarely noticed by other authors publishing in this area.
The issue described may in the future contribute to improving the efficiency of the automation, deep learning, denoising and underwater environmen.
This paper presents an improved denoising autoencoder based on DANAE, which is able to recover Kalman Filter IMU. Further tests could make this method suitable for real-time applications on navigation tasks.
For a better clarification, please edit your paper as follows: 1. Extend the text of manuscript (example introduction or conclusion) to concrete results in the world and in Europe, - Improve the quality of the paper by presenting the results of publications of researchers and experts that are registered in the world databases (wos). They are specifically these: Experimental investigations of a highly maneuverable mobile omniwheel robot, Navigation control and stability investigation of a mobile robot based on a hexacopter equipped with an integrated manipulator and
Integration of Inertial Sensor Data into Control of the Mobile Platform. Thanks. 2. figures 1 should be contrasting and readable,
3. conclusions and future work should be extended to contain practical applications based on research described in this paper - expand references, 4. highlight the course of dependencies/relations in figure No. 4 - 7, the red color is indistinct,
5. the paper should be read by a native english speaker.
I recommend publishing the post after the proposed modifications.

Author Response

Thank you for your comments. Please find attached our response. Thank you.

Reviewer 2 Report

This paper proposed a neural network to be used for denoising the output of a Kalman filter in the process of estimating the output of an underwater vehicle. The paper is overall well-written and interesting, the contribution is stated clearly in the Introduction and the literature taken into account is relevant. However in the Reviewer's opinion a major issue is the validation section, that has to be strengthened because it is not convincing.
- the first dataset involves data that do not come from an underwater environment. The authors should give a more detailed explanation on the reason why training the system on this data is meaningful for an underwater application
- the Kalman filter implemented for comparison is too poorly tuned (the initialization of the covariance matrices as identity matrices do not make much sense), so the comparison might be unfair
- the validation is performed only on few data. it would be more meaningful to validate the NN on a real underwater environment to assess the validity of the approach
- looking at the figures, I do not see much difference between applying the NN or a simple low-pass filter to the KF output. Can the author highlight the benefit of their approach in this sense?

Minor comments:
- lines 146-148: I do not understand how this matlab example can be helpful for the reader. In my opinion this can be removed
- line 155: the euler angles do not measure the attitude, they express it
- line 264: "the six intermediate ones extrapolated from the KF", please explain better
- line 300: "with a precision down to 0.5 mm". What about the orientation error? the position error is not useful in your application
- All the figures: please rewrite all the labels, uniform all the axes for a more intuitive comparison

Author Response

(The authors gave the same response as above.)

Round 2

Reviewer 2 Report

The authors have properly addressed all the issues I have raised in the first review.